# Discovery of Nosiheptide, Griseoviridin, and Etamycin as Potent Anti-Mycobacterial Agents against *Mycobacterium avium* Complex

**DOI:** 10.3390/molecules24081495

**Published:** 2019-04-16

**Authors:** Kanji Hosoda, Nobuhiro Koyama, Akihiko Kanamoto, Hiroshi Tomoda

**Affiliations:** 1Department of Microbial Chemistry, Graduate School of Pharmaceutical Sciences, Kitasato University, Tokyo 108-8641, Japan; hosodak@pharm.kitasato-u.ac.jp; 2Medicinal Research Laboratories, School of Pharmacy, Kitasato University, Tokyo 108-8641, Japan; 3OP BIO FACTORY Co., Ltd., 5-8 Suzaki, Uruma-shi, Okinawa 904-2234, Japan; akihiko.kanamoto@opbio.com

**Keywords:** nontuberculous mycobacteria, *Mycobacterium avium* complex, griseoviridin, viridogrisein, etamycin, nosiheptide, antimycobacterial activity, microbial product, marine-derived actinomycete

## Abstract

*Mycobacterium avium* complex (MAC) is a serious disease mainly caused by *M. avium* and *M. intracellulare.* Although the incidence of MAC infection is increasing worldwide, only a few agents are clinically used, and their therapeutic effects are limited. Therefore, new anti-MAC agents are needed. Approximately 6600 microbial samples were screened for new anti-mycobacterial agents that inhibit the growth of both *M. avium* and *M. intracellulare*, and two culture broths derived from marine actinomycete strains OPMA1245 and OPMA1730 had strong activity. Nosiheptide (**1**) was isolated from the culture broth of OPMA1245, and griseoviridin (**2**) and etamycin (viridogrisein) (**3**) were isolated from the culture broth of OPMA1730. They had potent anti-mycobacterial activity against *M. avium* and *M. intracellulare* with minimum inhibitory concentrations (MICs) between 0.024 and 1.56 μg/mL. In addition, a combination of **2** and **3** markedly enhanced the anti-mycobacterial activity against both *M. avium* and *M. intracellulare*. Furthermore, a combination **2** and **3** had a therapeutic effect comparable to that of ethambutol in a silkworm infection assay with *M. smegmatis*.

## 1. Introduction

*Mycobacterium avium* complex (MAC) is a serious disease mainly caused by *M. avium* and *M. intracellulare*. In recent years, the number of MAC patients has exceeded the number of tuberculosis patients [1,2]. Although clarithromycin (CAM), ethambutol (EB), and rifampicin (RFP) are used to treat MAC infection, their therapeutic effects are limited, and patients must take them for a long time (over one year). In addition, mycobacteria resistant to these clinically used agents are increasingly reported [3,4,5]. Therefore, it is important to develop new drugs for MAC infection. Accordingly, we screened for microbial compounds that were effective for MAC infection by selecting microbial culture broths that inhibit the growth of both *M. avium* and *M. intracellulare*. We found two marine-derived actinomycetes *Streptomyces* spp. OPMA1245 and OPMA1730. A thiopeptide identified as nosiheptide (**1**) [6] was isolated from the culture broth of the former strain, and two streptogramins identified as griseoviridin (**2**) [7] and etamycin (viridogrisein) (**3**) [8] were isolated from the culture broth of the latter strain (Figure 1). These compounds were previously isolated, but we found that they had anti-MAC activity and described the anti-mycobacterial activities and therapeutic effects in an in vivo mimic silkworm infection model.

## 2. Results

### 2.1. In Vitro Anti-Mycobacterial Activities of ***1–3***

The anti-mycobacterial activities of **1–3** on *M. avium, M. intracellulare, M. smegmatis*, and *M. bovis* were evaluated according to a liquid microdilution method [9] (Table 1). Among the three compounds, **1** had the most potent anti-mycobacterial activity against *M. avium, M. intracellulare*, and *M. bovis* with the lowest minimum inhibitory concentrations (MICs; 0.024, 0.024, and 0.012 μg/mL, respectively) followed by those of **3**, and compound **2** had modest MICs (1.56, 1.56, and 6.25 μg/mL, respectively). The MICs of **1** and **3** were comparable to those of clinically used agents, CAM and RFP. Furthermore, all the compounds had higher MICs against *M. smegmatis* than those against the other mycobacteria listed in Table 1.

### 2.2. Combination Effect of Streptogramins

Noeske et al. reported that streptogramins (combination of group A, dalfopristin, and group B quinupristin) have synergistic effects on the growth of gram-positive bacteria such as *Enterococcus faecium*, *Staphylococcus aureus*, and *Streptococcus pyogenes* [10]. In our case, **2** belongs to group A, and **3** belongs to group B. Therefore, we investigated whether a combination of **2** and **3** had a synergistic inhibitory effect on *M. avium* and *M. intracellulare* according to the checkerboard method [11]. For instance, the MIC of **2** on *M. avium* and *M. intracellulare* was reduced from 1.56 μg/mL to 0.019 μg/mL and from 1.56 μg/mL to 0.039 μg/mL in combination with **3** (0.007 μg/mL to 0.031 μg/mL), maximally yielding 80-fold and 40-fold potentiation, respectively (Table 2). Furthermore, the fractional inhibitory concentration (FIC) indexes in these combinations were below 0.5, which suggests that their interaction was synergistic. Furthermore, to confirm the synergistic activity of **2** and **3** against the three mycobacteria, the ratios of **2** and **3** (1:1 and 3:7) were set up based on a previous study using streptogramins. As shown in Table 3, the combinations of these ratios were synergistic against *M. avium*, *M. intracellulare*, and *M. bovis* BCG with reduced MICs ranging between 0.02 and 0.09 μg/mL.

### 2.3. Therapeutic Effect of ***1*** and ***2/3*** in a Silkworm Infection Model

We previously established a mycobacterial silkworm infection assay with *M. smegmatis* [12]. *M. smegmatis* is a nonpathogenic bacterium. However, when *M. smegmatis* was injected to silkworms at a higher cell number (10^7^ CFU/larva) and the injected silkworms were incubated at a higher temperature (37 °C), they all died within 50 h. In addition, clinically used anti-TB drugs, such as isoniazid, rifampicin, and ethambutol, showed a therapeutic effect in this silkworm assay [12]. Therefore, we evaluated **1** and a **2**+**3** mixture (in a 1:1 ratio) in this assay model. After being infected with *M. smegmatis* (*n* = 5), all silkworms died within 41 h. Under this condition, **1** showed a moderate therapeutic effect in the silkworm infection assay by prolonging survival time several hours (Figure 2). On the other hand, **2**/**3** increased the survival of silkworms in a dose-dependent manner, and silkworms survived for longer than 14 h at the maximal dose of **2**/**3** (Figure 2). Furthermore, the 50% effective dose (ED_50_) values of these compounds 41 h after infection were 35.4 µg/larva (Table 4) indicating that **2**/**3** (ED_50_, 35.4 µg/larva) had better efficacy than that of **1** and comparable efficacy to that of EB. In addition, **1** and **2/3** did not have toxicity on silkworms for at least 48 h.

## 3. Discussion

In the present study, approximately 6600 microbial culture broth samples were screened for new anti-MAC compounds using *M. avium* and *M. intracellulare.* Our screening results indicated that 0.06% of samples were active against only *M. avium*, 5.7% were active against only *M. intracellulare*, and 5.4% were active against both *M. avium* and *M. intracellulare*. We focused on samples active against both mycobacteria to obtain agents that may be clinically useful and selected samples with particularly potent activity. The selected strains were re-cultured, and the re-cultured broths that had reproducible activity were selected (0.03%).

Two culture broths from marine-derived actinomycetes OPMA1245 and OPMA1730 had potent activity against both mycobacteria. We isolated compounds from the culture broths and found three active compounds, nosiheptide (**1**), griseoviridin (**2**), and etamycin (viridogrisein) (**3**; Figure 1). These antibiotics were discovered in the 1950s and 1970s, but little was known about their anti-mycobacterial activity [13,14]. We found that they had potent activity against two MAC mycobacteria (Table 1). In an in vitro assay, **1** was the most potent followed by **3** and **2**. As reported by Noeske et al., a combination of group A (containing an unsaturated lactam ring) and group B (containing a cyclohexadepsipeptide) of streptogramin antibiotics has a synergistic effect on the growth of Gram-positive bacteria [10]. Based on this finding, **2** (group A) and **3** (group B) were mixed in a 1:1 ratio to investigate anti-MAC activity. We demonstrated the synergistic activity of **2**/**3** on two mycobacteria (Table 3) and the in vivo efficacy in silkworms infected with *M. smegmatis* (Figure 2). Intriguingly, **1** was the most potent in vitro on the four mycobacteria (Table 1), but **2**/**3** had better efficacy than that of **1** in the silkworm infection assay. The in vitro potency of **2**/**3** on *M. smegmatis* (MIC, 6.25 µg/mL) was 8-fold weaker than that of EB (MIC, 0.78 µg/mL), whereas **2**/**3** had the same in vivo therapeutic effects (ED_50_, 35.4 µg/mL) in the silkworm assay (Table 4).

The therapeutic potency of these compounds has been reported. For example, **1** and **3** has therapeutic efficacy in the mice infection model with methicillin-resistant *S. aureus* [13,15]. However, chiefly due to unfavorable physico-chemical properties, **1** has been used previously only for applications in farm animals, mainly as a feed additive [16].

Hamamoto et al. previously reported that the ED_50_/MIC ratio of a compound is an index of drug potential, and the ratio is typically below 10 for clinically useful antibiotics [14]. As shown in Table 4, the ratio of **2**/**3** was below 10, indicating that **2**/**3** is a potential anti-MAC drug. Combination therapy of dalfopristin (group A) and quinupristin (group B) is clinically used for the treatment of vancomycin-resistant *Enterococcus faecium* [17]. Therefore, streptogramin combination therapy may be effective to treat MAC infection. Since effective antimicrobial therapies for MAC infection are limited, these antibiotics should be re-evaluated as lead compounds for anti-MAC drugs.

## 4. Materials and Methods

### 4.1. Assay for Anti-Mycobacterial Activity

*The M. smegmatis* M341 and *M. bovis* BCG Pasteur used were laboratory strains. *M. avium* JCM15430 and *M. intracellulare* JCM6384 were purchased from the Riken BioResource Research Center (Ibaraki, Japan). Anti-mycobacterial activities against these four strains were evaluated by the liquid microdilution method according to a previously established method [12,18].

*M. avium* and *M. intracellulare* were routinely cultured for 72 h and 96 h, respectively. The suspensions were diluted with Middlebrook 7H9 broth (Becton Dickinson, MD, USA) containing 0.5% Tween 80 (MP Biomedicals, Santa Ana, CA, USA) and 10% ADC enrichment (Sigma Aldrich, MO, USA) to adjust to 4.0 × 10^6^ CFU/mL. Diluted mycobacteria (95 μL) were added to each well of a 96-well microplate (Corning, NY, USA) with or without a test sample or drug (in 5 μL MeOH or DMSO) and incubated at 37 °C for 120 h. Then, MTT (3-(4,5-dimethyl-2-thiazolyl)-2,5-diphenyl-2H tetrazolium bromide, Sigma Aldrich) reagent (5.5 mg/mL, 5 µL H_2_O) was added to each well, and the cells were incubated for 16 h. Lysis buffer (40% *N*,*N*-dimethylformamide, 20% SDS, 2% CH_3_COOH, 95 μL) was added to each well to dissolve the formazan pigment. The absorbance was measured at a wavelength of 570 nm using a UV plate reader (BioTek, VT, USA) to evaluate the growth of the mycobacterium. The MIC was defined as the lowest drug concentration that inhibited mycobacterium growth by 90%.

### 4.2. Isolation of **1**–**3**

The actinomycete strain *Streptomyces* sp. OPMA1245 was isolated from marine sediment collected in Okinawa prefecture, Japan. The 7-day-old culture broth of OPMA1245 (3.0 L) was extracted with an equal volume of EtOAc. After the EtOAc extracts were filtered, the EtOAc layer was concentrated to yield 204.3 mg of a yellow solid material. This material was dissolved in a small volume of DMSO, applied to an ODS column (10 g, 1.0 cm × 20 cm), and eluted stepwise with 100% H_2_O, 20%, 40%, 60%, 70%, 80%, and 100% CH_3_CN (80 mL each). Compound **1** was eluted with 60% CH_3_CN. The whole eluate was concentrated in vacuo to dryness to yield a yellow solid material (110.0 mg). Compound **1** was finally purified by preparative high-performance liquid chromatography (HPLC) under the following conditions: Column, PEGASIL ODS SP100 (i.d. 20 mm × 250 mm); mobile phase, 40-min gradient from 30% CH_3_CN to 70% CH_3_CN; flow rate, 6.0 mL/min; detection, UV at 210 nm. Under these conditions, **1** was eluted as a peak with a retention time of 31 min, yielding pure **1** (6.8 mg) as a yellow powder.

The actinomycete strain *Streptomyces* sp. OPMA1730 was isolated from marine algae collected in Okinawa prefecture, Japan. The 8-day-old culture broth of OPMA1730 (4.5 L) was extracted with an equal volume of EtOAc. After the EtOAc extracts were filtered, the EtOAc fraction was concentrated to yield 376.5 mg of a yellow solid material. This material was dissolved in a small volume of MeOH, applied to an ODS column (20 g, 2.0 cm × 20 cm), and eluted stepwise with 100% H_2_O, 20%, 30%, 40%, 50%, 60%, 70%, and 100% CH_3_CN (120 mL each). The 30% CH_3_CN fraction containing **2** was concentrated to give a yellow solid material (53.4 mg). Compound **2** was purified by preparative HPLC: Column, PEGASIL ODS SP100 (i.d. 20 mm × 250 mm); mobile phase, 23% CH_3_CN; flow rate, 6.0 mL/min; detection, UV at 210 nm. Under these conditions, **2** was eluted as a peak with a retention time of 37 min, yielding pure **2** (12.5 mg) as a white powder. The 50% CH_3_CN fraction containing **3** was concentrated to give a yellow solid material (54.5 mg). Compound **3** was also purified by preparative HPLC: Column, PEGASIL ODS SP100 (i.d. 20 × 250 mm); mobile phase, 65% CH_3_CN; flow rate, 6.0 mL/min; detection, UV at 210 nm. Under these conditions, **3** was eluted as a peak with a retention time of 20 min, yielding pure **3** (30.8 mg) as a pale yellow solid.

Their structures were elucidated by various spectroscopic analyses including NMR experiments (Appendix A). Based on reference data described in the literature [19,20,21], compounds **1**, **2**, and **3** were identified as nosiheptide, griseoviridin, and etamycin (viridogrisein), respectively (see details in the Appendix A).

### 4.3. Measurement of FIC Indexes

The synergistic effects of the combination of **2**/**3** were evaluated according to the checkerboard method [10]. Briefly, two-fold serial dilution samples (5 μL) of **2** were added to rows 1 to 11 in a 96-well plate. Then, serial two-fold dilution samples (5 μL) of **3** were added to columns A to G in the 96-well plate. In addition, those of **2** were added to row 12, and those of **3** were added to column H to measure each MIC. After drying the samples and re-dissolving in MeOH (5 μL), mycobacterial culture (95 μL) was added to each well and incubated at 37 °C for 120 h. MICs of **2** and **3** in combination were determined. For the wells of the 96-well plate that corresponded to each MIC, an FIC index was calculated for each well with Equation (1) as below [11]:

**Equation (1)**. FIC index:FIC index = FIC_2_^a^ + FIC_3_^a^ = (C_2_^b^/MIC_2_^c^) + (C_3_^b^/MIC_3_^c^)(1)
where a: FIC of **2** and **3**; b: C_2_ and C_3_ are the MICs of **2** and **3** in combination, respectively; and c: MIC_2_ and MIC_3_ are the MICs of **2** and **3** alone, respectively

### 4.4. Silkworm Infection Assay with M. smegmatis

The assay for silkworm infection with *M. smegmatis* was performed according to our established method [11]. Hatched silkworm larvae (Ehime Sansyu, Ehime, Japan) were raised by feeding Silk Mate 2S (Nihon Nosan Kogyo, Kanagawa, Japan) in an incubator at 27 °C until the fourth molting stage. On the first day of fifth-instar larvae, silkworms (*n* = 5) were fed Silk Mate 2S. On the second day, *M. smegmatis* (2.5 × 10^7^ CFU/larva in 50 μL Middlebrook 7H9 broth) was injected into the hemolymph through the dorsal surface of the silkworm using a disposable 1-mL syringe with a 27-G needle (TERUMO, Tokyo, Japan). A sample solubilized in 50 μL of 10% DMSO was injected to the hemolymph within 1 h of infection with *M. smegmatis*. All *M. smegmatis*-infected silkworms died within 41 h when no sample was administered. After sample injection, the number of silkworms that survived was counted at the indicated time until 70 h. The survival rate at the indicated dose of each sample was calculated when all *M. smegmatis*-infected silkworms without sample injection had died (41 h). The ED_50_ values (at 41 h after infection) were calculated according to a previous method [12,15,22].

## Figures and Tables

**Figure 1 molecules-24-01495-f001:**
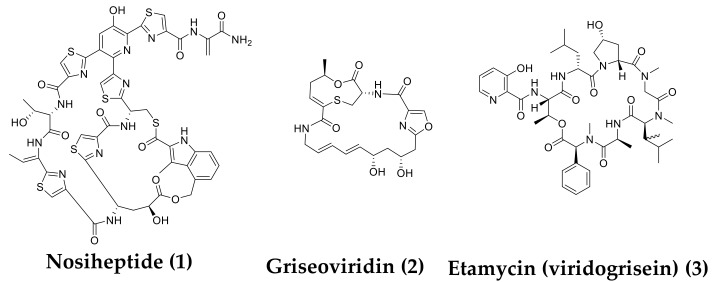
Structures of **1**–**3**.

**Figure 2 molecules-24-01495-f002:**
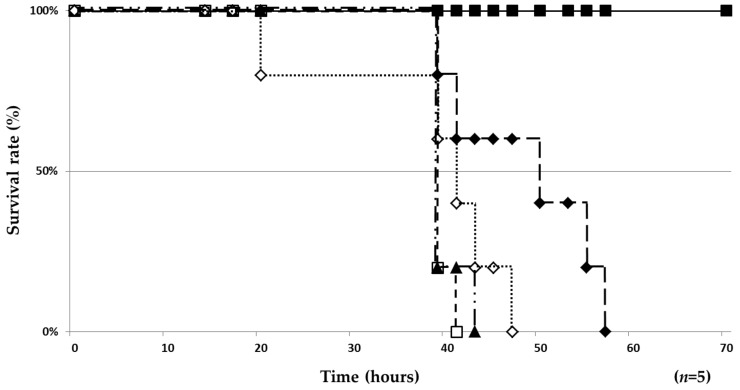
Therapeutic effects of **1** and **2**/**3** in the silkworm infection assay with *M. smegmatis.* Dose: ◊ 50 µg/larva for **1**. ▲ 12.5 and ◆ 50 µg/larva for **2**/**3.** ☐ control (no drug). ■ no infection with *M. smegmatis*. Experiments were performed twice to confirm reproducibility.

**Table 1 molecules-24-01495-t001:** Minimum inhibitory concentration (MIC) values of 1–3 against mycobacteria.

Test Microorganism	MIC (µg/mL)					
	1	2	3	CAM	RFP	EB
*Mycobacterium avium* JCM15430	0.024	1.56	0.097	0.19	0.78	12.5
*Mycobacterium intracellulare* JCM6384	0.024	1.56	0.19	0.024	0.012	3.12
*Mycobacterium smegmatis* M341	6.25	>100	25	15.6	1.56	0.78
*Mycobacterium bovis* BCG Pasteur	0.012	6.25	0.78	0.12	0.012	1.56

**Table 2 molecules-24-01495-t002:** MICs of **2** in combination with **3** against *Mycobacterium avium* and *M. intracellulare.*

In Combination with 3	MIC of 2 (FIC Index *) Against	
	*M. avium* JCM15430	*M. intracellulare* JCM6384
(µg/mL)	(µg/mL)	
0	1.56	1.56
+0.007	0.15 (0.078)	0.078 (0.093)
+0.015	0.039 (0.046)	0.078 (0.15)
+0.031	0.019 (0.070)	0.039 (0.26)

* The fractional inhibitory concentration (FIC) index was calculated by Equation (1). Synergistic: FIC index ≤ 0.5, Additive: 0.5 < FIC index ≤ 1.0, No synergy: 1.0 < FIC index ≤ 2.0.

**Table 3 molecules-24-01495-t003:** MICs of **2**+**3** mixtures against mycobacteria.

Test Microorganism	MIC (µg/mL)	
	2+3 (1:1)	2+3 (3:7)
*M. avium* JCM15430	0.024	0.097
*M. intracellulare* JCM6384	0.048	0.097
*M. smegmatis* M341	6.25	6.25
*M. bovis* BCG Pasteur	0.024	0.012

**Table 4 molecules-24-01495-t004:** ED_50_ values of **1** and **2**/**3** in the silkworm infection assay with *M. smegmatis.*

Test Compound	ED_50_ (µg/larva) ^1^	MIC (µg/mL)	ED_50_/MIC
**1**	>50	6.25	>8
**2**/**3**	35.4	6.25	5.6
**EB**	35.4	0.78	45

^1^: 50% effective dose.

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
