# Peer review of "Discovery of Nosiheptide, Griseoviridin, and Etamycin as Potent Anti-Mycobacterial Agents against *Mycobacterium avium* Complex"

_molecules, 2019, doi:10.3390/molecules24081495_

Round 1
Reviewer 1 Report
Marine products were evaluated as potent antimycobacterial agents against Mycobacterium avium complex. A thiopeptide nosiheptide, two streptogramins (peptidolactone antibiotics) griesoviridine (no longer available) and etamycin (more preferred name than the used name viridogrisein)produced by Streptomyces griseoviridus are described as potent drugs for MAC infection. Both synergistic streptogramin antibiotics are very important in treating infections of many multi-drug resistant microorganisms. Authors used a mycobacterial silkworm infection assay with Mycobacterium smegmatis to determine marine products activity.
I have found only few discrepancies in the manuscript.
Please, use the INN name for the etamycin.
More recent information (citations) about resistant mycobacteria are needed in Introduction. Citations 3 and 4 ara comming from 2004, resp. 2007. I suppose, the situation with resistant mycobacteria should be quite different in 2019.
There is no other information about the therapeutic potency of studied compounds. Thiopeptides nosiheptide, for example have potent activity against various bacterial pathogens, but it posses low water solubility and poor resorption from the gastrointestinal tract. It is used as a feed additive in the growth of poultry and hogs to promote growth and general health. Please, add such important information in the article.
I would like to regard highly the preparation of the evaluated manuscript, on other side there is a lack of other important information, therefore I don’t recommend the publication of the manuscript evaluated in this form. I recommend the publication of the manuscript evaluated after minor revision from the reasons noted above.
Author Response
Thank you for your suggestion.
We agree with you and have incorporated this suggestion throughout our paper.
Point 1: Please, use the INN name for the etamycin.
Response 1: According to your suggestion, we changed viridogrisein to etamycin. Accordingly, the title was also changed.
Point 2: More recent information (citations) about resistant mycobacteria are needed in Introduction. Citations 3 and 4 ara comming from 2004, resp. 2007. I suppose, the situation with resistant mycobacteria should be quite different in 2019.
Response 2: According to your suggestion, we added a recent good review about the drug-resistant mycobacteria on reference [5].
Reference;
5) Adelman, M.H.; Addrizzo-Harris, D.J. Management of nontuberculous mycobacterial pulmonary disease. Curr. Opin. Pulm. Med. 2018, 24, 212-219, 10.1097/MCP.0000000000000473.
Point 3: There is no other information about the therapeutic potency of studied compounds. Thiopeptides nosiheptide, for example have potent activity against various bacterial pathogens, but it posses low water solubility and poor resorption from the gastrointestinal tract. It is used as a feed additive in the growth of poultry and hogs to promote growth and general health.
Response 3: Thank you for your thoughtful suggestion. Accordingly, we added the information and reference about these compounds in lines 119 to 122 in the text.
“The therapeutic potency of these compounds has been reported. For example, 1 and 3 has therapeutic efficacy in the mice infection model with methicillin-resistant S. aureus [13,15]. However, chiefly due to unfavorable physico-chemical properties, 1 has been used previously only for applications in farm animals, mainly as a feed additive [16].”
Reference;
16) Benazet, F.; Cartier, J.R. Effect of Nosiheptide as a Feed Additive in Chicks on the Quantity, Duration, Prevalence of Excretion, and Resistance to Antibacterial Agents of Salmonella typhimurium; on the Proportion of Escherichia coli and other Coliforms Resistant to Antibacterial Agents; and on Their Degree and Spectrum of Resistance. Poult. Sci. 1980, 59, 1405-1415, 10.3382/ps.0591405.
Reviewer 2 Report
This is an interesting and obviously properly performed study for which I suggest only a few additions/amendments:
I would like to know from where the authentic samples of 1 and 3 derived?
For the reader not too familiar with anti-mycibacterial assays it might be feasible to have a short explanatory statement that the assay in silk worms was performed with M. smegmatis as the only non-pathogenic strain.
In the tables I would prefer the decimal points at the same position in each column.
I could not find any citation of reference 3 in the Supplementary material. Why do you cite this reference there?
Author Response
Thank you for your suggestion.
We agree with you and have incorporated this suggestion throughout our paper.
Point 1: I would like to know from where the authentic samples of 1 and 3 derived?
Response 1: We obtained the authentic sample 1 from Prof. Hans-Dieter Arndt, School of Pharmacy, Friedrich-Schiller-Universität Jena (reference [20]). So we added Acknowledgments for him to providing the sample.
We did not obtain the authentic sample 3. The structure of 3 was elucidated by comparison with reported data of viridogrisein (reference [19]).
Point 2: For the reader not too familiar with anti-mycibacterial assays it might be feasible to have a short explanatory statement that the assay in silkworms was performed with M. smegmatis as the only non-pathogenic strain.
Response 2: We have tested the points raised by the reviewer in detail in reference [12]. Anyway, we added the following sentences in lines 77 to 81.
“M. smegmatis is a nonpathogenic bacterium. But, when M. smegmatis was injected to silkworms at a higher cell number (107 CFU/larva) and the injected silkworms were incubated at a higher temperature (37 °C), they all died within 50 hours. In addition, clinically used anti-TB drugs, such as isoniazid, rifampicin, and ethambutol, showed therapeutic effect in this silkworm assay [12].”
Point 3: In the tables I would prefer the decimal points at the same position in each column.
Response 3: According to your suggestion, we fixed it.
Point 4: I could not find any citation of reference 3 in the Supplementary material. Why do you cite this reference there?
Response 4: Thank you for your indication. We deleted reference [3] and changed reference [4] to reference [3].
Reviewer 3 Report
A paper entitled "Discovery of nosiheptide, griseoviridin, and viridogrisein as potential anti-mycobacterial agents aganist Mycobacterium avium complex" is submitted to Molecules for further reviewing and publication. All the isolates are well known antibiotics, due to the novelty on Chemistry is not high to attract reader's interesting. I don't recommend that this submission is acceptable for publication with its form. This paper should be submitted to another specific journal focus on microorganisms or the journal such as 'Biological & Pharmaceutical Bulletin".
Author Response
Thank you for your suggestion.
Comment and Suggestion: A paper entitled "Discovery of nosiheptide, griseoviridin, and viridogrisein as potential anti-mycobacterial agents aganist Mycobacterium avium complex" is submitted to Molecules for further reviewing and publication. All the isolates are well known antibiotics, due to the novelty on Chemistry is not high to attract reader's interesting. I don't recommend that this submission is acceptable for publication with its form. This paper should be submitted to another specific journal focus on microorganisms or the journal such as 'Biological & Pharmaceutical Bulletin".
Response : You have raised an important point; All the isolates are well known antibiotics. Due to the novelty on chemistry it is not high to attract reader's interesting. However, we believe that new biological activities of well know compounds become very important. As we described in this manuscript, much attention has been paid on MAC infection and new drugs for the treatment of MAC. Recently, study of drug repositioning or drug-refinding is actively on-going.
In this sence, we believe that our findings about anti-MAC activity of known compounds are important to many readers.
Round 2
Reviewer 3 Report
As my comment in the first round. Novelty in chemistry is not high to attract reader's interesting. Although, biological activity for the isolates play an important role to enhance the quality of this submission. The editorial office have to decide whether to accept this submission.